# Morpho-Anatomical, Physiological and Biochemical Adjustments in Response to Heat and Drought Co-Stress in Winter Barley

**DOI:** 10.3390/plants12223907

**Published:** 2023-11-20

**Authors:** Emmanuel Asante Jampoh, Eszter Sáfrán, Dorina Babinyec-Czifra, Zoltán Kristóf, Barbara Krárné Péntek, Attila Fábián, Beáta Barnabás, Katalin Jäger

**Affiliations:** 1Biological Resources Department, HUN-REN Centre for Agricultural Research, 2462 Martonvásár, Hungary; emmanuel.jampoh@atk.hun-ren.hu (E.A.J.); eszter.safran@atk.hun-ren.hu (E.S.); babinyec.dorina@atk.hun-ren.hu (D.B.-C.); pentek.barbara@atk.hun-ren.hu (B.K.P.); fabian.attila@atk.hun-ren.hu (A.F.); barnabas.beata@gmail.com (B.B.); 2Doctoral School of Horticultural Sciences, MATE Hungarian University of Agriculture and Life Sciences, 2100 Gödöllő, Hungary; 3Doctoral School of Biology, Institute of Biology, ELTE Eötvös Loránd University, 1053 Budapest, Hungary; 4Department of Plant Anatomy, ELTE Eötvös Loránd University, 1053 Budapest, Hungary; kristof.zoltan@ttk.elte.hu

**Keywords:** microgametogenesis, phenology, photosynthetic pigments, photosynthesis, ultrastructure, yield

## Abstract

This study aimed to investigate the combined effect of high temperatures 10 °C above the optimum and water withholding during microgametogenesis on vegetative processes and determine the response of winter barley genotypes with contrasting tolerance. For this purpose, two barley varieties were analyzed to compare the effect of heat and drought co-stress on their phenology, morpho-anatomy, physiological and biochemical responses and yield constituents. Genotypic variation was observed in response to heat and drought co-stress, which was attributed to differences in anatomy, ultrastructure and physiological and metabolic processes. The co-stress-induced reduction in relative water content, total soluble protein and carbohydrate contents, photosynthetic pigment contents and photosynthetic efficiency of the sensitive Spinner variety was significantly greater than the tolerant Lambada genotype. Based on these observations, it has been concluded that the heat-and-drought stress-tolerance of the Lambada variety is related to the lower initial chlorophyll content of the leaves, the relative resistance of photosynthetic pigments towards stress-triggered degradation, retained photosynthetic parameters and better-preserved leaf ultrastructure. Understanding the key factors underlying heat and drought co-stress tolerance in barley may enable breeders to create barley varieties with improved yield stability under a changing climate.

## 1. Introduction

Greenhouse gases released due to industrialization accumulate in the atmosphere and result in global warming. The increasing average temperature of the atmosphere, ocean and land results in climate change with high-temperature extremes and harsh alteration in rainfall patterns in terms of temporal and spatial distribution, resulting in extended drought periods [1,2]. The incidence of transient abnormal rise in temperature above the normal physiological range [3,4] and frequent severe drought periods have been predicted to increase [5].

Through evolution, plants as sessile organisms have developed appropriate strategies with dynamic responses at the morphological, physiological and biochemical levels to enable their survival and reproduction even under suboptimal conditions. Plant sensitivity to adverse environmental conditions is majorly dependent on the severity of the stress as well as plant species, genotype and developmental stage. Frequently occurring transient heatwaves and drought episodes pose a major threat to plant production and food security as it leads to a reduction in the development, yield and quality of crops cultivated under suboptimal environmental conditions [6,7,8]. Drought triggers the reduction of plant morphology and alters physiological traits [9,10,11]. Prolonged heat or drought periods are not limited to arid and semi-arid regions but also increasingly affect agricultural production in temperate regions. Cultivation areas are already increasingly subjected to intermittent and terminal drought and/or heat stress, which generally have the strongest effects on plants during the reproductive phase of development [12,13]. 

Most studies often consider the effects of heat or drought on plant development per se; however, the impact of stressors is generally exacerbated when interacting [14,15,16]. These co-stresses are severely detrimental or even fatal to plants as compared to their individual effect [14,17,18,19,20]. Plant response to single abiotic factors applied individually cannot justify the uniqueness of plant response to a concurrent combination of different abiotic factors [21,22,23]; for a review, see [24]. 

Improvement in plant production and yield safety is required to meet the food demand of the increasing human population [25], even under a changing climate. It is thus important to understand the mechanisms contributing to adaptation/tolerance to heat and drought co-stress typically occurring in the field during generative development in both model systems and crop plants to collect agronomically relevant data to validate the detrimental consequences of climate extremes predicted for the future. An increasing amount of available literature is concerned with the negative effect of heat and drought (HD) co-stress applied during flower initiation and meiosis or anthesis and grain development on reproductive processes and the yield of cereals [26,27]. However, there is a dearth of information available on the effect of HD co-stress during microgametogenesis [28], even though the formation of a functional vegetative cell and sperm cells within the male gametophyte, the development of the inner pollen wall and deposition of starch reserves needed for pollen tube growth and successful fertilization, all take place during this important developmental period. 

Barley, belonging to the grass family (Poaceae), is the fourth most produced cereal in the world, covering 51.6 million hectares with an average yield of 3.04 t ha^−1^ and a total production of 157 million tons in 2020 [29]. Barley is the cereal crop with the widest and most diverse geographical distribution grown under varied agro-climatic conditions and used as food, fodder and as a raw material for malt production [30]. 

The hypothesis tested in this paper was that a short period of HD co-stress can markedly affect the anatomy and physiological and biochemical processes of barley flag leaves, the organs that are known to be responsible for the major part of carbohydrate production [31], and assimilate allocation to developing generative organs [32]. In this study, the effect of HD co-stress on the physiological and biochemical status and anatomy of the flag leaves was evaluated in terms of water content, osmotic adjustment, total soluble protein content, photosynthetic pigment content, water-soluble carbohydrate content, starch content, proline content, glycine betaine content, as well as the structure and ultrastructure of mesophyll cells. The chlorophyll *a* fluorescence and gas exchange parameters of both genotypes were assessed from the flag leaves during co-stress treatment and recovery.

## 2. Results

### 2.1. Stress-Induced Changes in Plant Phenology

Compared to their respective controls, the duration of grain filling was markedly shortened by 9 days in HD co-stress tolerant Lambada and by 12 days in HD sensitive Spinner plants.

### 2.2. Effect of HD Co-Stress on Growth and Yield

Under control conditions, compared to the Lambada variety, the plant height and peduncle length of the Spinner genotype were 31% and 34% greater, respectively. HD co-stress caused a significant 12% and 21% reduction in plant height, mainly due to the 40% and 46% decline in the peduncle length, as compared to the controls in Lambada and Spinner, respectively (Table 1). Neither of the genotypes showed signs of leaf rolling because of HD co-stress (Figure 1); however, flag leaves of the Spinner variety were rather wilted. Moreover, HD caused yellowing and senescence of lower leaves in the Spinner variety (Figure 1d).

Both genotypes experienced significant losses in their measured yield constituents after HD co-stress was applied at microgametogenesis; however, Spinner produced lower values (Figure 2). HD co-stress induced a drop in the grains per productive tiller in both Lambada and Spinner by significant (*p* ≤ 0.05) 27% and 54% (Figure 2a), as well as reduced the thousand-grain weight in both varieties by 24% and 32%, respectively (Figure 2b). HD co-stress subsequently triggered a significant reduction in plant production in Lambada and Spinner genotypes of up to 43% and 57%, respectively (Figure 2c). The HI was significantly reduced in Lambada by 27% and Spinner by 41% (Figure 2d). According to the effect of HD co-stress on yield components of barley genotypes, Lambada was further considered a more tolerant variety, and Spinner was considered a sensitive variety.

### 2.3. Combined Heat and Drought Stress-Induced Alterations in Water Relations

The mean volumetric water content (VWC) of the soil during HD co-stress applied for 5 days dropped from 37.1% to 14.1%, which was equal to 12 kPa and 373 kPa soil water potential, respectively. Treatment did not result in a significant decline in the relative water content (RWC) of the flag leaves of the Lambada genotype (Figure 3a). In contrast, HD co-stress resulted in a significant 39% reduction in the RWC of the Spinner variety as compared to its control. There was a moderate correlation observed between the RWC recorded at anthesis and plant height and peduncle length measured at harvest (Table 2).

### 2.4. Genotype-Dependent Differences in Osmotic Adjustment

The osmotic adjustment (OA) of the HD co-stressed Lambada and Spinner plants was −0.600 MPa and −0.917 MPa, respectively, which indicates a significant difference in the accumulation of osmolytes between the two genotypes (Figure 3b). 

### 2.5. Heat and Drought Co-Stress Reduced Total Soluble Protein and Photosynthetic Pigment Contents

The amount of total soluble proteins in the control flag leaves of both Lambada (33.4 mg g^−1^ DW) and Spinner (48.4 mg g^−1^ DW) genotypes was significantly reduced due to the applied HD treatment by 31% and 63%, respectively, as compared to their respective control counterparts (Figure 4a). The concentration of the photosynthetic pigments was measured in terms of the Chl_a_, Chl_b_, Chl_a+b_ and Car contents in the flag leaves, and the ratios of Chl_a_/Chl_b_ and Car/Chl were calculated. The chlorophyll contents of the control plants varied with the genotypes. Compared to the total pigment contents of the Spinner flag leaves (10.1 mg g^−1^ DW), those of Lambada (16.8 mg g^−1^ DW) were significantly lower under control conditions (Figure 4b–d). On a dry weight basis, Chl_a_, Chl_b_ and Chl_a+b_ contents underwent a significant decrease under HD co-stress conditions in both Lambada and Spinner flag leaves by 16%, 19% and 16% and 60%, 62% and 61%, respectively. Based on the acquired values, the reduction in these pigments was much more substantial in the plants of the Spinner genotype, the lower leaves of which turned yellow at the end of the 5-day treatment (Figure 1d). Nevertheless, the decrease in HD-induced total Chl contents (−16%, −60%) was less than the decrease in P_n_ (−46%, −80%). HD co-stress resulted in a 5% non-significant increase in Car contents of Lambada flag leaves despite inducing a 53% significant decline in the Spinner variety (Figure 4e). HD co-stress significantly increased the chlorophyll *a*/*b* ratio in the flag leaves of both Lambada and Spinner by 4% and 6%, respectively (Figure 4f). The Car/Chl ratio was increased by a significant 21% and 15% in Lambada and Spinner varieties, respectively (Figure 4g). The HD-induced reduction of Chl_a_ and Chl_b_ contents was in a strong positive correlation with the decline of the plant RWC (Table 2).

### 2.6. Stress-Induced Decline in Chlorophyll a Fluorescence and Leaf Gas Exchange Parameters

There was no significant difference in the F_v_/F_m_ ratio among the genotypes under control conditions (mean = 0.8), nor between the F_v_/F_m_ values of the Lambada and Spinner plants measured at flowering (Figure 5a). In our opinion, the statistically significant 2% decrease in the F_v_/F_m_ ratio of the HD co-stressed plants of the Spinner variety was not caused by the physiological changes triggered by the 5-day stress treatment but by the low standard deviation of the measured values. HD co-stress had no effect on the ETR of the genotypes (Figure 5b). The decline of the Φ_PSII_ was measured in Spinner plants on the first day of HD co-stress (Figure 5c). Simultaneous HD had no effect on the qP during the treatment; however, at the end of the regeneration period, the parameter increased significantly (Figure 5d) in both genotypes. Compared to the respective controls, a genotype-independent HD co-stress-triggered increase in NPQ was observed both along the treatment and regeneration period (Figure 5e). In contrast to the F_v_/F_m_ ratio, there was a significant difference between the mean P_n_ values of Lambada (8.88 ± 0.26 µmol CO_2_ m^−2^ s^−1^) and Spinner plants (6.34 ± 0.72 µmol CO_2_ m^−2^ s^−1^) grown under control conditions. Such a significant difference was measured between the P_n_ values of the varieties during the experiment (Figure 6a). The P_n_ of both genotypes decreased significantly by the end of the treatment (S5); however, compared to the decline in Lambada (49%), the reduction in the net photosynthesis of Spinner plants (84%) was significantly greater. Although the P_n_ of the tolerant variety returned to the initial value on the first day of regeneration, only a gradual increase in this parameter was detected in the sensitive genotype during regeneration. Despite P_n_, no differences in mean g_s_, C_i_, E, WUE or ICE were observed between the genotypes under control conditions. Compared to the Lambada variety, the g_s_ of the sensitive genotype was significantly lower during both the stress treatment and regeneration period (Figure 6b). The highest reduction in the g_s_ of the HD co-stressed plants was observed at S5, with Lambada having relatively higher values. Compared to S5, g_s_ measured during the regeneration was significantly higher in both genotypes; however, the stomatal conductance of Spinner leaves never reached the initial value measured at S1. Compared to S1, a gradual decrease of C_i_ was observed in HD co-stressed Lambada leaves with the inflection point on the last day of the treatment (stomatal limitation of photosynthesis; Figure 6c). In the Spinner variety, a sudden increase in C_i_ was observed at S5, indicating the predominance of non-stomatal limitations to photosynthesis in this genotype. After a remarkable drop at S3-5, the E was increased during the regeneration in both varieties (Figure 6d). Compared to S1, the WUE of both genotypes arose significantly on the third day of treatment (Figure 6e). At the same time, the WUE of Lambada plants showed a 4.46-fold increase at S5 compared to its initial value at S1. The WUE of the Spinner variety returned to the initial value at S4. The ICE of the sensitive genotype increased significantly on the 4th day of treatment (S4), which suddenly dropped below its initial value measured at S1 on the following day (S5; Figure 6f). A temporary nine-fold increase in the same parameter of the tolerant Lambada plants was observed on the last day of the HD co-stress. The F_v_/F_m_, P_n_, g_s_ and E were in a significantly strong and moderate correlation with RWC, respectively (Table 2).

### 2.7. HD-Induced Alterations in Proline, Glycine Betaine, Starch and Total Water-Soluble Carbohydrate Contents

There were no genotype-dependent differences either in proline or in glycine betaine (GB) accumulation under control conditions between the genotypes (Figure 7a,b). HD induced a significant (*p* ≤ 0.05) increase of proline accumulation in both varieties; however, compared to the 156% increase in Lambada, a vast 2027% increase in Spinner flag leaves occurred (Figure 7a). Interestingly, proline contents were negatively correlated to all traits examined but Car (Table 2). Strong or very strong significant interactions were found between proline and RWC, F_v_/F_m_, P_n_, Chl_b_, Car, GB, GPT or grain production. In contrast to proline, the effect of HD on deposited GB was different: it was not affected in Lambada plants, but a significant 57% reduction of GB content was observed in Spinner flag leaves (Figure 7b). GB had a significant positive correlation with all traits but WUE, Car, proline and HI. There was no genotype-dependent difference in the amount of TSC between the two varieties under control conditions (Figure 7c). In contrast, HD co-stress induced a 78% increase in the TSC content of Lambada and a 25% decrease in Spinner flag leaves. There was a strong negative significant correlation between C_i_ and TSC and a strong positive significant correlation between WUE or ICE and TSC (Table 2). Compared to Lambada, the spectrophotometric assay shed light on higher rates of starch accumulation in the flag leaves of the control Spinner plants; however, HD induced a sharp 95% decline of starch deposition in both varieties (Figure 7d). No significant correlation was found between TSC and starch content of the flag leaves at the end of HD co-stress (Table 2). 

### 2.8. High Temperature and Water Deprivation Triggered Alterations in Leaf Anatomy

Under control conditions, compared to the Lambada plants, the flag leaves of the Spinner variety were significantly (*p* ≤ 0.05) thicker by 15% (Table 2; Figure 8a,b). Nevertheless, the leaf thickness of HD co-stressed Lambada plants was significantly increased by 23% (Figure 8c) and decreased by 18% in Spinner (Figure 8d) because of the expansion or shrinkage of sub-stomatal cavities, respectively. Independent of the genotype, HD co-stress had neither an effect on the size of the bulliform and mesophyll cells nor the number of chloroplasts per mesophyll cell (Table 2).

### 2.9. Combined Stress Altered Starch Granule Accumulation in Mesophyll Chloroplasts

In agreement with the spectrophotometric analysis of leaf starch content (Figure 7d), HD co-stress triggered alterations in the starch accumulation in mesophyll cells (Figure 8e–h). The chloroplasts of the control flag leaves accumulated starch in both genotypes at anthesis (Figure 8e,f); however, the number of starch granules was significantly higher in the Spinner variety. In contrast, no histochemically labeled starch granules were present in HD co-stressed leaves of either of the studied barley genotypes (Figure 8g,h). 

### 2.10. HD-Triggered Ultracellular Changes in Mesophyll Cells

Mesophyll cells in control Lambada flag leaves showed native ultrastructure. The cytoplasm contained photo-respiratory complexes consisting of chloroplasts, mitochondria and peroxisomes (Figure 9a). The matrix of the mitochondria was electron-dense. The kidney-shaped chloroplasts possessed intact outer membranes and well-developed intermembrane system arranged in a plane, which was composed of closely packed granas and connecting stroma thylakoids. Starch granules were rarely present, and only a low number of plastoglobules were distributed in the stroma. Spinner mesophyll cells with a healthy ultrastructure contained well-developed, starch-containing chloroplasts (Figure 9b); however, the thylakoid system of the chloroplasts was less regular than in the Lambada variety. The tonoplast of the central vacuole was intact, and the peripheral cytoplasm contained a large amount of mitochondria with visible cristae. HD co-stress altered the shape of the Lambada chloroplasts: in many cases, the organelle took on an amoeboid shape and protrusions formed at the ends (Figure 9c). The plane of the thylakoid membranes was twisted, so thylakoids with different orientations were visible. The granas persisted, and the swelling of the lumen was not typical in them compared to the stroma thylakoids. The envelope membranes of the plastids and mitochondria remained intact. Plastoglobules were barely visible in the plastids. HD co-stress had a more significant effect on the ultrastructure of the Spinner variety (Figure 9d). The chloroplasts were swollen, became amoeboid and contained numerous plastoglobules. The plastids often formed extensions with large depressions and lobes. These extensions were not penetrated by the irregularly arranged and randomly oriented thylakoid system. In contrast to the Lambada variety, the lumen of the grana thylakoids expanded, in some cases, to an extreme extent. Swollen mitochondria with disorganized cristae were present in the vacuolated cytoplasm. Waves appeared on the cell wall, which indicates that the cells were shrinking. No starch granules were present in the chloroplasts of either of the HD-treated barley varieties (Figure 9c,d).

## 3. Discussion

Although microgametogenesis is a critical period of sexual plant reproduction, little is known about the effect of heat and drought co-stress during this developmental phase on the vegetative and generative processes of barley that affect yield constituents and, ultimately, yield. In our study, HD co-stress shortened the period of grain filling in both Lambada and Spinner plants by 20% and 31%, respectively. Similarly, previous studies demonstrated a reduction in the days to physiological maturity in wheat when subjected to combined HD stress at microgametogenesis and heading, respectively [21,28,33]. Reduced life cycle length has a significant impact on the functionality of the generative organs and cells, overall assimilate accumulation, biomass and the length of the effective grain filling period, resulting in an overall loss in the seed weight in many crop species [34].

Analysis of agronomic traits showed that combined HD stress during microgametogenesis took a heavy toll on both varieties. Co-stress-triggered inhibition of peduncle elongation and extrusion from the pseudostem was the main contributor to the reduction in plant height in both Lambada and Spinner genotypes, however, with higher restrictions in the latter sensitive variety. Both plant height and peduncle length depend on the genotype and environmental conditions [35,36]. Similarly to our results, HD co-stress caused a reduction in plant height in wheat [35] and inhibited peduncle growth in rice by trapping spikelets inside the flag leaf sheath [36]. Aside from the flag leaf, non-foliar organs like the spike and the exposed part of the peduncle are photosynthetically active and produce and convey photosynthates into caryopses, thereby making a crucial contribution to grain growth [31,37]. It is obvious why HD co-stressed barley plants with only partially or non-exposed peduncles produced grains with significantly lower TGW. Reduced grain number in both Lambada and Spinner spikes was attributed to photosynthetic constraints and impairment of water relations due to HD co-stress during microgametogenesis, which might have caused strict limitations on the availability of assimilates to developing male gametophytes and so triggered the partial failure of fertilization. Grain filling is predominantly dependent on sugars being translocated from the leaves, leaf sheaths, stems and bracts, as well as those synthesized in the maturing caryopsis [38]. As with chickpeas and lentils [39,40,41], in the present study, reduced area of photosynthesizing organs (senescent leaves, non-exserted peduncles), impairment in photosynthesis during regeneration and shortened life cycle, all cut the production of sugars and starch in barley leaves and assimilate translocation into developing seeds. Limited availability of assimilates caused a reduction in the grain set and TGW and, thus, grain production. In contrast to a previous study that suggested a correlation between plant height and yield [35], we found a negligible connection between these two traits (Table 2). In line with a previous work [42], we found a moderate correlation between peduncle length and yield. As HD co-stress drove barley plants to produce caryopses in reduced number and size, albeit the vegetative tissues were almost fully developed (except the peduncles) by the beginning of the treatment, the HI was also significantly cut. Overall, co-stress reduced the yield components of both varieties; however, the impact of HD on the stress-sensitive Spinner plants was more severe.

The protein contents were significantly reduced under HD co-stress conditions in the flag leaves of both genotypes, in line with previous studies made on lentils, wheat and maize [33,43,44,45]. During senescence, the enhanced degradation of protein contents in the leaves facilitates their remobilization to seeds. Although plants can metabolize proteins and lipids as alternative respiratory substrates, the respiration of proteins is less efficient than that of carbohydrates. At the same time, under certain adverse conditions, this can represent an important alternative energy source for the cell [46].

In our study, HD-tolerant Lambada plants exhibited a noticeable adaptive feature where the number of chloroplasts in the mesophyll cells and the concentration of photosynthetic pigments in the flag leaves cultivated under control conditions were lower as compared to the HD-sensitive Spinner variety. Comparably, HD co-stress caused the downregulation of the chlorophyll content in barley; however, the stress was applied at the seedling stage [47]. Maintaining a lower concentration of chlorophylls in leaf tissues can be considered a protection mechanism against PSII damage, and it seems to be a general feature of the genetic adaptation of plants to harsh environments [48,49]. Our results suggest that the lower chlorophyll content of the Lambada variety decreased leaf light absorbance, which reduced the damaging excess heating effect of solar radiation in HD co-stressed plants with almost completely inhibited transpiration. Plant wilting and premature leaf senescence (yellowing) of the lower leaves were visible as early as the 4th day of the HD treatment in the sensitive Spinner plants. During leaf senescence, the translocation of salvaged nutrients to be used by emerging leaves and reproductive organs takes place in addition to chlorosis, a remarkable phenomenon associated with the deterioration of chlorophyll and chlorophyll–protein complexes [50]. The foliar content of chlorophylls and Car and especially their ratios (Chl_a_/Chl_b_, Car/Chl) constitute a sensitive indicator of the physiological status of plants during their development and acclimation to diverse environmental stimuli and stresses [51,52]. A decline in chlorophyll concentrations during stress can be attributed to both the inhibition of chlorophyll biosynthesis and the denaturing of chlorophyll molecules due to the disruption of chloroplast membranes [41]. HD-induced degradation of the photosynthetic pigment contents resulted in decreased photosynthesis that led to impaired carbon fixation in the HD co-stress-sensitive Spinner variety. Similarly to our results, reduced chlorophyll content-induced lowered light-harvesting capacity was reported in drought-stressed plants [10,53,54]. In the present study, the Chl_a_/Chl_b_ ratio was significantly increased with a concomitant decrease in the total chlorophyll concentration in both barley genotypes during HD co-stress treatment, indicating a higher proportion of Chl_b_ degraded or converted to Chl_a_ [55]. As both photosystem I and photosystem II reaction centers are devoid of Chl_b_, the increase in Chl_a_/Chl_b_ ratio reflects the reduction in the size of light-harvesting complex II [56], which decreases light-harvesting capacity. In contrast to the significant decline in treated Spinner plants, the carotenoid content of the Lambada genotype did not vary with treatment, indicating a persistent photoprotection and scavenging of toxic oxygen species in the chloroplasts. The elevation in the Car/Chl ratio that often reflects the reduction of photosynthetic apparatus and, hence, plant photosynthetic capacity in response to stresses [52,57] was also revealed in both barley varieties. 

We assume that the reduced conductance of HD co-stressed Spinner stomata measured throughout the experiment resulted not from the breed’s lower stomatal density but from greater stress sensitivity, as no genotype-dependent differences were measured between the g_s_ values of the controls. However, the conductance of the Lambada stomata was reduced gradually to a quarter to the 4th day of treatment, and the CO_2_ content of the intercellular spaces was sufficient to maintain the P_n_ at 76% of the initial value. In contrast, the intercellular CO_2_ was used in the leaves with closed stomata of the Spinner variety, and the net photosynthesis was reduced by 44% on the 4th day of HD co-stress. Probably as a result of maintaining the ability to keep the stomata open and sustaining CO_2_ fixation under stress conditions, the net photosynthesis of the tolerant variety decreased to only 51% of the control to the end of treatment. In contrast, the plants of the Spinner variety, which had a higher loss in the chlorophyll concentration, closed their stomata to a greater extent and, as a result of the severely reduced photosynthesis, the intercellular CO_2_ content increased six times compared to the previous day’s value. S5 also recorded reduced E as compared to S1 in both Lambada and Spinner genotypes. Similarly, HD co-stress reduced P_n_ in lentils and olives, which correlated with a reduction in g_s_ [4,41]. Photosynthesis is positively related to CO_2_ assimilation, and the reduction in photosynthesis is attributed to increased photorespiration and a reduction in the RuBisCo activase enzyme [4]. In the present experiment, the g_s_ and E were significantly reduced by HD co-stress in both genotypes, and the decline of both parameters significantly correlated with a reduction in RWC. This is in line with previous findings in *Arabidopsis thaliana*, where high temperature and a water deficit induced the closure of stomatal pores to minimize water loss [17,58]. A greater ability to sustain an enhanced photosynthetic capacity was observed in the Lambada genotype, as revealed by higher P_n_, g_s_, C_i,_ WUE and ICE values in S5 in comparison with Spinner. The higher P_n_, g_s_ and E of the tolerant genotype on the last day of co-stress treatment indicates the strategy of protection by transpirational cooling to alleviate thermal damage from leaf overheating. It is likely that the larger aboveground surface area of the drought-sensitive variety, which, compared to the tolerant genotype, was 20 cm and 10 cm taller under control and co-stress conditions, respectively, could be responsible for the more significant reduction in RWC. A reduced E in both barley genotypes, as observed in our study, could be linked to a reduced g_s_, similarly reported in maize [40]. Carbon uptake became limiting in Spinner plants in regards to stomatal closure, leading to increased leaf C_i_ and altered plant carbon balance. During recovery, Lambada exhibited a relatively higher P_n_ from R1–R5, showing fully recovered photosynthetic capacity. The ability to sustain a stronger photosynthetic capacity and less oxidative damage during drought has usually been associated with fast recovery after re-watering [5]. The enhanced photosynthetic capacity of Lambada during R1–R5 is also related to relatively improved g_s_ and sustained C_i_ during recovery. In contrast, HD co-stress resulted in a lower recovery rate of P_n_, g_s_ and E of Spinner plants. Slow photosynthetic recovery was confirmed in the heat-stressed leaves of *Phaseolus vulgaris* to be the result of the reduced capacity for photosynthetic electron transport [59].

In the present experiment, compared to the sensitive variety, Lambada plants showed greater RWC at the end of the HD co-stress treatment in contrast to lower OA. Heat or drought stress is commonly associated with the accumulation of osmoprotectants, which is known to be an efficient drought tolerance mechanism that helps to maintain plant functions even under limited water availability [60]. Sugars, glycine betaine and amino acids are among the most common compatible solutes [61], which can be accumulated in high concentrations without hindering normal physiological functions [62]. Our present study recorded no genotype-dependent differences in the amount of TSCs between the studied genotypes under control conditions. However, with the retained P_n_ and high ICE, the TSCs were significantly increased in the flag leaves of the HD-tolerant Lambada plants due to HD co-stress, and the carbohydrate content of Spinner leaves was reduced. This tolerance-dependent variation of TSC contents in barley is in line with HD-induced changes in reducing sugar contents described in lentil genotypes with different stress tolerance [41]. The TSCs play an important role in maintaining the overall structure and growth of plants, and they can maintain OA during stress conditions [63]. TSC accumulation may be sensitive to environmental stressors such as water deprivation or high temperature, which decrease the efficiency of photosynthesis and limit the TSC supply to sink organs [64]. Sugar starvation in the floral organs is the underlying factor in the reproductive failure of a rice cultivar sensitive to the combination of drought and heat [26].

GB is an osmolyte synthesized in the chloroplasts of plants and can protect the enzymes and membranes against heat [65] or drought stress [66]. During our experiments, no genotype-dependent differences in GB accumulation between the genotypes were recorded under control conditions. Although GB accumulation is generally correlated with stress tolerance [67], the varieties in the present study were not characterized by an excessive accumulation of GB under HD co-stress: stagnation was observed in the case of Lambada, and a sharp reduction in the case of Spinner plants. 

Heat and drought co-stress usually results in the excessive accumulation of reactive oxygen species (ROS), which are partially reduced forms of molecular oxygen (Mittler 2017) [68]. ROS function both as signal transduction molecules and also as toxic by-products of aerobic metabolism, causing drastic cellular damage [28,68]. Hence, to scavenge ROS [69], plants have evolved enzymatic and non-enzymatic antioxidant defense strategies like the synthesis of proline by the GS/GOGAT cycle [70] and its accumulation to avoid oxidative stress injuries [11,71]. Free proline is known to accumulate in response to several abiotic stresses and has multifunctional roles [72]. Proline metabolism is involved in the regulation of intracellular redox potential, the storage and transfer of energy and the reduction of power [73]. In contrast to the moderate increase of proline accumulation in HD-tolerant Lambada plants (156%), it was drastically elevated in the drought-sensitive variety up to 21-fold as compared to the control, despite having an overall stress response. As with our results, some authors also detected higher proline accumulation in susceptible cultivars of barley [74], sorghum [75], potato [76] and cassava [77].

The leaves of many plant species either fold or twist into tubes during drought as part of their defense strategy to alter leaf cell turgor or decrease transpiration and overheating [78]. Bulliform cells are situated in the upper epidermis and, due to water loss, cause transverse rolling of the leaf lamina along the mid-axis in some Gramineae species such as *sorghum*, rice, maize and wheat [79,80,81]. Neither HD-triggered leaf rolling nor a reduction in the plane area of the bulliform cells occurred in barley plants in the present study, although the RWC of the Spinner leaves dropped by 39%, and withering of the plants was observable. HD-tolerant Lambada flag leaves responded to simultaneous HD with increased thickness, as opposed to the flag leaves of the drought-sensitive Spinner variety, which became thinner through the shrinkage of sub-stomatal cavities. Comparably, lower water availability induced a higher proportion of sub-stomatal cavities and leaf thickness in *Sorghum* [79], probably to improve leaf capacity to preserve water while increasing the plant’s ability to take up CO_2_. In contrast to a previous study [80], the mesophyll cell area available for chloroplasts was not closely associated with the photosynthetic capacity of barley leaves. Moreover, the chloroplast number per mesophyll cell was not associated with photosynthetic capacity. Compared to Spinner plants, the photosynthetic activity of Lambada containing 25% fewer chloroplasts was higher under both control and HD co-stress conditions. No significant HD-induced reduction in the chloroplast number in either of the studied barley genotypes was recorded, which indicates retained integrity of the chloroplast outer membranes, in contrast to mesophyll cells in grapevine leaves under heat stress [81].

Mesophyll chloroplasts in the flag leaves accumulated primary starch in the leaves of both barley genotypes under control conditions; however, both structural and biochemical assessments of starch contents showed higher starch accumulation in the flag leaves of the Spinner variety. In contrast, independent of the genotype, compared to the respective controls, transitory starch in HD co-stressed mesophyll cells was reduced by 94.8%. In most plant species, 40–50% of the photoassimilates produced during the light period are transiently stored in the chloroplasts as primary starch [82]. Degradation of transitory starch in leaves typically occurs during the following night to provide the plant with energy and carbon in the absence of light or during the light period under unfavorable conditions that allow carbon to be reallocated from osmoprotectants that promote osmotic adjustment and stabilize proteins, scavenge free radicals and act as signals that refine stress responses [83]. 

To our knowledge, there is no information available on the effect of HD co-stress during generative development on the ultrastructure of barley flag leaves. In the present study, water deprivation for 5 days combined with a high temperature 10 °C above the optimum during microgametogenesis triggered disturbances in the ultrastructure of the photosynthetic organelles; however, the HD co-stress tolerant variety maintained the internal structure of the chloroplasts to a greater extent. 

In summary, we have shown that barley genotypes with contrasting stress tolerance were heavily compromised when subjected to terminal heat and drought co-stress during microgametogenesis. The simultaneous HD markedly altered water relations, photosynthetic ability and carbohydrate metabolism, resulting in stunted plants with reduced yield. However, compared to the sensitive Spinner variety, apart from its weak starch deposition in the mesophyll chloroplasts and lower proline accumulation in the flag leaves, the tolerant Lambada genotype was characterized by higher values in all studied physiological and biochemical traits as well as better preserved ultrastructure during and at the end of HD co-stress. Here, we show that HD co-stress tolerance of the Lambada variety might be associated with lower initial leaf chlorophyll contents, the relative resistance of photosynthetic pigments to HD-triggered degradation and high F_v_/F_m_, g_s_ and, thus, P_n_ during the treatment and the 5-day long regeneration period. These parameters could be important selection indicators for plants treated under HD stress. Moreover, despite their inherently lower plant height and lower aboveground plant mass, compared to the sensitive genotype, Lambada plants were able to produce and transfer more assimilates to the developing caryopses after the HD co-stress period due to their better preserved physiological and metabolic processes that resulted in higher grain number, TGW, yield and HI. A comprehensive analysis of the mechanisms underlying the high photosynthetic function with a low level of starch accumulation of the HD tolerant Lambada genotype with lower photosynthetic pigment content, as well as the exploration of the effect of HD co-stress on the functionality of the barley tapetum and male gametophytes, require further studies.

## 4. Materials and Methods

### 4.1. Plant Material

Two two-rowed winter barley (*Hordeum vulgare* L.) genotypes, Lambada (Regina × Rifle, RAGT UK, 2001) and Spinner (G84095-3043 × Velvet, Lantmännen, Sweden, 1999), with contrasting drought stress tolerance, were used in the experiment. Both genotypes, included in the genome-wide association studies panel composed of 882 barley accessions [84], were provided by The James Hutton Institute, Scotland, United Kingdom. 

### 4.2. Experimental Design

Each plant (n = 100 per genotype and treatment) was planted into a pot containing 2 kg of a soil-sand-peat mixture (3:1:1, *v*/*v*/*v*) after 8 weeks of vernalization at a temperature of 4 °C. Plants were grown in growth chambers (Conviron, Winnipeg, MB, Canada) under controlled environmental conditions using the T1 climatic program [85] with 16 h of day and 8 h of darkness until the mid-uninucleate (MU) stage of microspore development. The maximum photosynthetic photon flux density (PPFD) during the experiment was 300 μmol m^−2^ s^−1^ at a plant level that was supplied between 9:30 h and 13:00 h. Irrigation was carried out regularly in the morning at a rate of 150 mL/pot/day during this period. The plants were randomly rearranged every week within growth chambers during the experiment to reduce border effects stemming from variations in light and temperature. The MU stage of the microspores in the anthers of 5 plants for each genotype and treatment was checked by acetocarmine staining (1% Carmine in 45% acetic acid) and the distance between the auricles of the flag leaf and the penultimate leaf was recorded. A further selection of plants was based on the auricle distance. Plants with microspores at the MU stage in the main spikes were selected daily in the morning and transferred into control or stress chambers. The well-watered (control) plants were grown using the T1 climatic program with 18 h of day and 6 h of darkness from the MU stage of microspore development to flowering. The optimum min/max night/day temperature and relative air humidity were 10/20 °C and 75%/65%, respectively. The plants were watered with 150 mL of water per pot and per day, and the soil’s mean volumetric water content (VWC) was 37.1%. HD co-stress was induced by total water withholding at 20/30 °C min/max (night/day) temperature (10 °C above the optimum night/day temperatures during microgametogenesis) and 30%/65% min/max (day/night) relative air humidity for 5 days from the MU stage until flowering (Z60; [86]). During the HD co-stress treatment, the VWC of the soil dropped below 15% to the end of the treatment and strictly coincided with flowering in both genotypes in all 3 independent biological repetitions. Samples were collected at flowering and 15 intact plants of each genotype and treatment were re-irrigated and grown to full maturity under control conditions. The days from the MU stage to physiological maturity were noted for each genotype and treatment. The physiological maturity (Z90) was considered reached when the peduncles of the spikes became yellow.

### 4.3. Determination of Yield Components

The morphological features (tiller number, spike number, plant height, peduncle length) of control and HD co-stressed plants (n = 15 per genotype and treatment) were recorded at full maturity (Z90). The canopies were harvested, the aboveground biomass was measured and the spikes were hand-threshed. The grain number and grain weight were counted and weighed, respectively, and mean values were calculated for each treatment and cultivar. The grain number per productive tiller (GPT) and thousand-grain weight (TGW) were calculated from the data. The harvest index (HI) was calculated as the ratio of the harvested grain dry weight to the total canopy dry weight.

### 4.4. Water Status Measurements

An HH2 moisture meter connected to an SM200 soil water sensor (Delta-T Devices Ltd., Cambridge, UK) was used to keep track of the VWC of the soil (n = 10 pots per genotype and treatment). The relative water content (RWC) of the flag leaves (n = 15 per genotype and treatment) was calculated using the following equation: $RWC\left(\%\right)=\frac{FW-DW}{SW-DW}\times 100$ [87], where FW is the fresh weight at excision, SW is the saturated weight after 24 h rehydration on distilled water at 25 °C in the dark and DW is the dry weight after oven drying for 24 h at 80 °C.

### 4.5. Osmotic Adjustment

The osmolality of the leaf sap from the control and treated samples (3 flag leaves per genotype at the end of the treatment) was measured using an Osmomat 3000 cryoscopic osmometer (Gonotech GmbH, Berlin, Germany). For the measurement at full turgor (Ψ_S_100), flag leaves were rehydrated with deionized water for 6 h at 4 °C in the dark. Values of osmolality were converted from mOsmol·kg^−1^ to the osmotic potential values using the formula: Ψ_S_ (MPa) = −c (mOsmol·kg^−1^) × 2.58 × 10^−3^. Osmotic adjustment (OA) was calculated as the difference of Ψ_S_100 at full turgor (RWC = 100%) between unstressed and stressed samples. 

### 4.6. Chlorophyll a Fluorescence and Gas Exchange Measurements

The chlorophyll *a* fluorescence of intact flag leaves was measured using a PAM-2000 pulse amplitude modulation chlorophyll fluorometer (Walz, Effeltrich, Germany) at the MU stage of development (1st day of treatment), at anthesis (5th day of treatment) and on the 5th day of regeneration. The minimal fluorescence yield (F_0_) of the dark-adapted samples with all the photosystem II (PSII) reaction centers open was determined at modulated light (<0.1 μmol m^−2^ s^−1^). The maximum fluorescence yield of the dark-adapted samples with all the PSII centers closed (F_m_) was determined by 0.8 s saturating pulses of 2000 μmolm^−2^s^−1^ photosynthetic photon flux density (PPFD). Potential/maximum photochemical quantum yield values (F_v_/F_m_, where F_v_ is the variable fluorescence of the dark-adapted samples defined as F_m_˗F_0_) were determined after a leaf adaptation period of 20 min in the dark. The fluorescence parameters determined in both the light-adapted and dark-adapted leaves allowed the calculation of the quantum yield of photochemical energy conversion in PSII (Φ_PSII_), apparent electron transport rate (ETR), the photochemical quenching parameter related to the fraction of open reaction centers in PSII (qP) and non-photochemical quenching parameter (NPQ) describing regulated dissipation of excess energy by the PamWin software version 1.24 (Walz, Effeltrich, Germany). Gas exchange analysis (n = 5 pots per genotype and treatment) was performed on the intact flag leaves during treatment (S1–S5) and the 5-day-long regeneration period (R1-R5) with an LI-6400 infrared gas analyzer (LI-COR, Lincoln, NE, USA). The parameters net photosynthetic rate (P_n_), stomatal conductance to water vapor (g_s_), intercellular CO_2_ concentration (C_i_) and transpiration rate (E) were measured. The water use efficiency (WUE) and the instantaneous ribulose 1,5-diphosphate carboxylase/oxygenase carboxylation efficiency (ICE) were calculated as P_n_/E and P_n_/C_i_, respectively.

### 4.7. Photosynthetic Pigment Analysis

The chlorophyll *a* (Chl_a_), chlorophyll *b* (Chl_b_) and carotenoid (Car) contents of the flag leaves (n = 15 per genotype and treatment) were determined [88]. For each treatment, weighed 5 cm^2^ flag leaves were taken and homogenized in liquid nitrogen, 2 × 500 μL 80% acetone was added to the samples and the homogenized mixture was centrifuged at 13,440× *g* for 15 min at 4 °C. The supernatant was analyzed for Chl_a_, Chl_b_, chlorophyll *a* + *b* (Chl_a+b_) and carotenoid contents using a DU^®^ 730 spectrophotometer (Beckman Coulter, Brea, CA, USA). Equations used to determine concentrations (μgml^−1^) of Chl_a_, Chl_b_, Chl_a+b_ and Car were as follows: Chl_a_ = (12.25 × A664) − (2.79 × A646), Chl_b_ = (21.50 × A646) − (5.1 × A664), Chl_a+b_ = (7.15 × A664) + (18.71 × A646), Car = (1000 × A470 − 1.82 × Chl_a_ − 85.02 × Chl_b_) 198^−1^. The ratios of Chl_a_/Chl_b_ and Car/Chl_a+b_ were calculated using the data.

### 4.8. Measurement of Glycine Betaine, Proline, Protein, Starch and Carbohydrate Contents

The glycine betaine content was measured according to the assay of Grieve and Grattan (1983) [89]. The concentration of proline and starch was determined by the ninhydrin method [90] and the perchloric acid KI-I2 method [91], respectively. The total soluble protein (TSP) level of the flag leaves (n = 15 per genotype and treatment) was measured using a UV spectrophotometer [92]. Calculation of the contents was performed by creating standard curves using standards and was expressed in mg g^−1^ DW or μM g^−1^ DW. The total soluble carbohydrate (TSC) content was measured based on the Anthrone method [93].

### 4.9. Histological Studies

Segments (n = 6 per genotype and treatment) from the central region of the flag leaves were cut and collected at anthesis, fixed in 60 mM Sorensen’s phosphate buffer (pH 7.2) containing 4% paraformaldehyde (*w*/*v*) overnight at 4 °C, dehydrated in a series of ethanol, gradually infiltrated with LR white acrylic resin (Ted Pella, Redding, CA, USA) and the resin was polymerized at UV light at −20 °C. Semi-thin sections (1 μm) were cut using an Ultracut-E microtome (Reichert-Jung, Heidelberg, Germany), stained with periodic acid-Schiff (PAS) and 1% Amido Black for polysaccharides and proteins, respectively, mounted in 50% glycerol containing 7% acetic acid and examined with a BX51 light microscope (Olympus, Tokyo, Japan). The cross-sectional areas of the bulliform cells, mesophyll cells and the number of chloroplasts (minimum n = 120 measurements per trait, genotype and treatment) were measured and counted using an Image-Pro Plus image analysis software version 7.0 (Media Cybernetics, Inc., Bethesda, MD, USA).

### 4.10. Transmission Electron Microscopy

Segments (1 mm^2^; n = 6 per genotype and treatment) were cut from the central regions of the flag leaves at anthesis and fixed with 2.5% (*v*/*v*) glutaraldehyde dissolved in 0.1 M Sörensen’s buffer (SB; pH 7.2) for 4 h and overnight at RT and 4 °C, respectively. Samples were postfixed with 1% OsO_4_ for 4 h, dehydrated through ascending ethanol series, embedded in Durcupan ACM (Merck, Darmstadt, Germany) resin according to the manufacturer’s instructions and cut using an Ultracut-E microtome (Reichert-Jung, Heidelberg, Germany). The ultrathin sections (70 nm) were mounted on 100-mesh nickel grids coated with Formvar (SPI-Chem, West Chester, PA, USA) and examined by a transmission electron microscope (Hitachi 7100) operating at 80 keV.

### 4.11. Statistical Analysis

All data were pooled means from the replicates, subjected to ANOVA (SSPS version 16.0, IBM Corp., Armonk, NY, USA), and the mean values were compared by Tukey’s multiple range test, taking *p* ≤ 0.05 as significant, to compare the differences between treatments and genotypes. Pearson’s correlation coefficient was used to deduce the relationships between the measured traits (SSPS for Windows, version 16.0, IBM Corp., Armonk, NY, USA). Correlation coefficients (r) were interpreted as follows: 1–0.9 absolute magnitude (AM) of very strong correlation, 0.89–0.7 AM of strong correlation, 0.69–0.4 AM of moderate correlation, 0.39–0.1 AM of weak correlation, 0.1–0 AM of negligible correlation [94]. Mean values and standard deviations are presented in tables and figures. The standard deviation (SD) of means and differences between treatments were compared pairwise at *p* ≤ 0.05 level of probability. 

## Figures and Tables

**Figure 1 plants-12-03907-f001:**
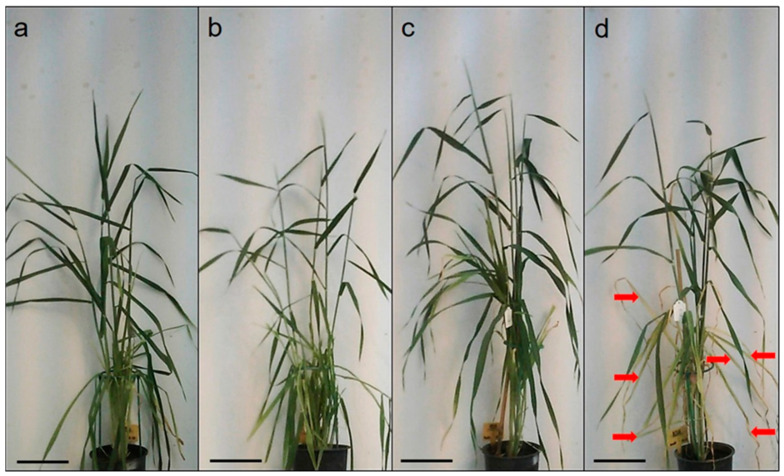
Control and heat and drought co-stressed (HD) barley plants at anthesis. (**a**) Control Lambada; (**b**) HD Lambada; (**c**) control Spinner; (**d**) HD Spinner (representative images). Arrow—yellowing, senescent leaf. Scale bar = 10 cm.

**Figure 2 plants-12-03907-f002:**
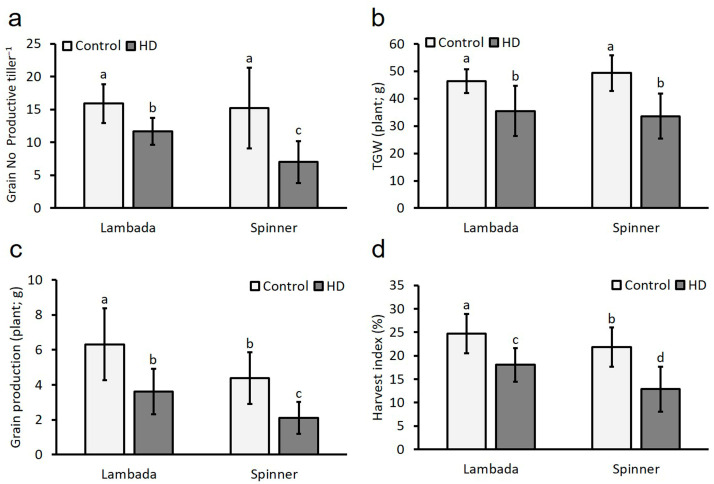
Yield components (grain number to productive tiller ratio (**a**); thousand-grain weight (**b**); grain production (**c**); harvest index (**d**) of control and HD Lambada and Spinner plants. TGW—thousand-grain weight. Mean values, along with standard deviations, are presented. In each histogram, different letters above columns indicate significant differences between means at the *p* ≤ 0.05 level of probability.

**Figure 3 plants-12-03907-f003:**
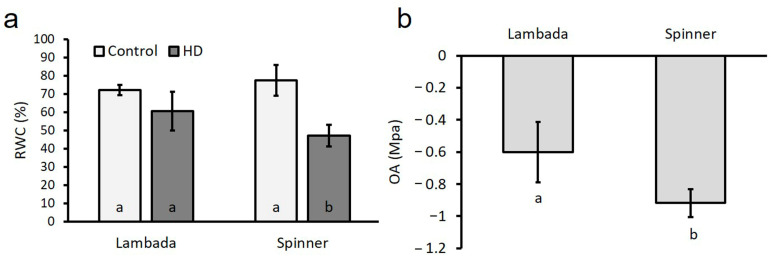
Relative water content (**a**) and osmotic adjustment (**b**) of control and HD barley flag leaves. OA—osmotic adjustment, RWC—relative water content. Mean values, along with standard deviations, are presented. In each histogram, different letters within columns indicate significant differences between means minimum at the *p* ≤ 0.05 level of probability.

**Figure 4 plants-12-03907-f004:**
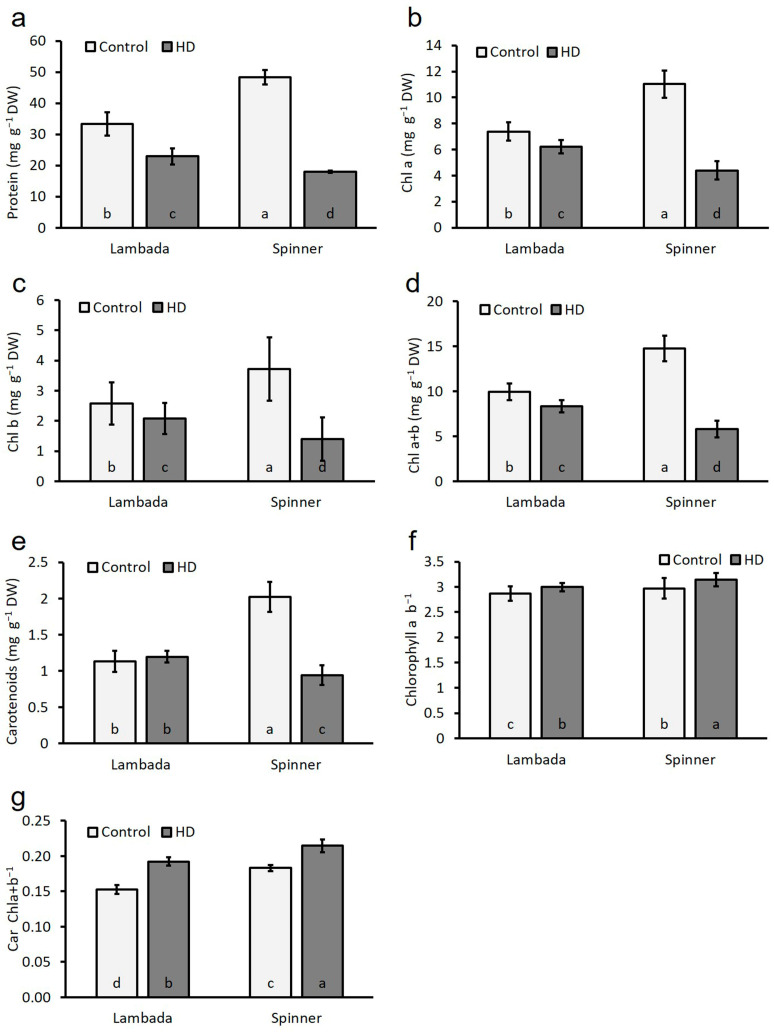
Total soluble protein content (**a**), photosynthetic pigment contents (**b**–**e**), chlorophyll *a* to chlorophyll *b* ratio (**f**), and carotenoids to chlorophyll *a + b* ratio (**g**) in the flag leaves of the control and HD Lambada and Spinner plants. Chl—chlorophyll, DW—dry weight. Mean values, along with standard deviations, are presented. In each histogram, different letters within columns indicate significant differences between means at the *p* ≤ 0.05 level of probability.

**Figure 5 plants-12-03907-f005:**
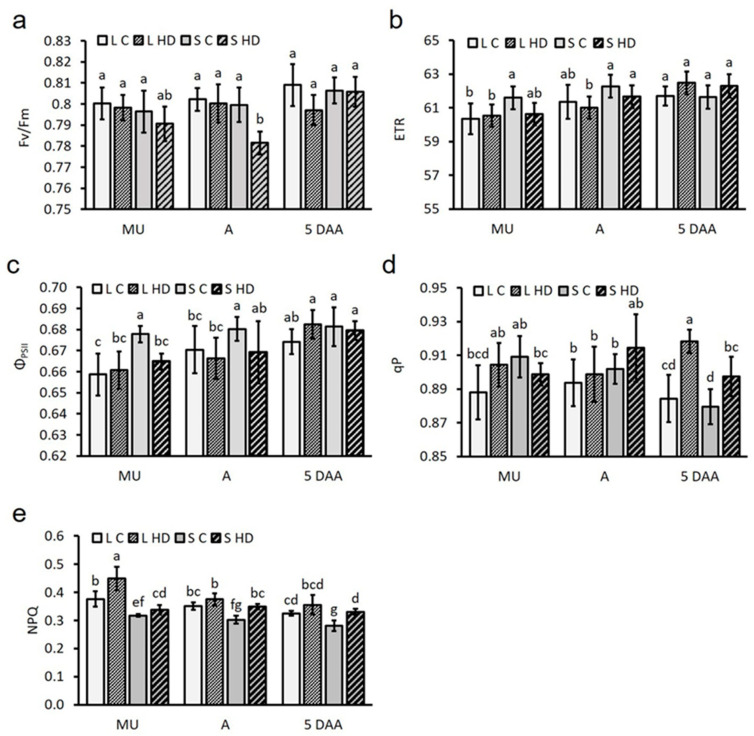
Maximum quantum yield of the photosystem II (**a**), relative electron transport rate at PSII (**b**), actual quantum yield of PS II (**c**), photochemical quenching coefficient (**d**) and non-photochemical quenching (**e**) measured 1 day after the mid-uninucleate stage of microspore development, at anthesis and 5 days after anthesis in barley. C—control, L—Lambada, R5—5th day of regeneration, S—Spinner, S1—1st day of HD co-stress treatment, S5—5th day of HD co-stress treatment. Mean values, along with standard deviations, are presented. In each histogram, different letters above columns indicate significant differences between means at the *p* ≤ 0.05 level of probability.

**Figure 6 plants-12-03907-f006:**
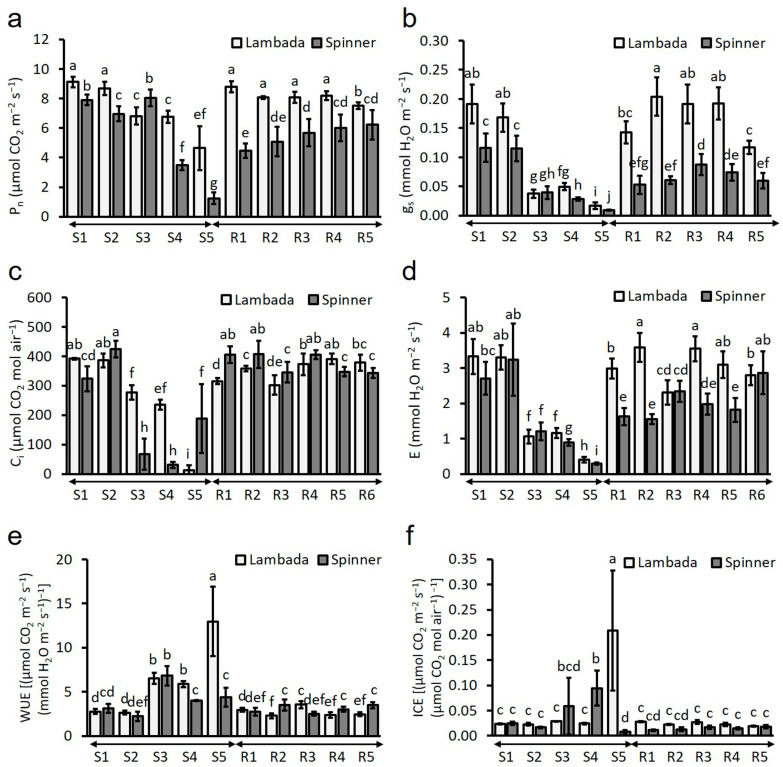
Measured time series of net photosynthesis (**a**), stomatal conductance (**b**), intracellular CO_2_ concentration (**c**), transpiration rate (**d**), water use efficiency (**e**) and instantaneous carboxylation efficiency (**f**) during the 5-day-long HD co-stress (S1–S5) and regeneration (R1–R5) periods in barley genotypes with different stress-susceptibility. C_i_—internal CO_2_ concentration in the sub-stomatal chamber, E—transpiration rate, WUE—water use efficiency, F_v_/F_m_—maximum quantum efficiency, g_s_—stomatal conductance, P_n_—CO_2_ assimilation rate. Mean values, along with standard deviations, are presented. In each histogram, different letters above columns indicate significant differences between means at the *p* ≤ 0.05 level of probability.

**Figure 7 plants-12-03907-f007:**
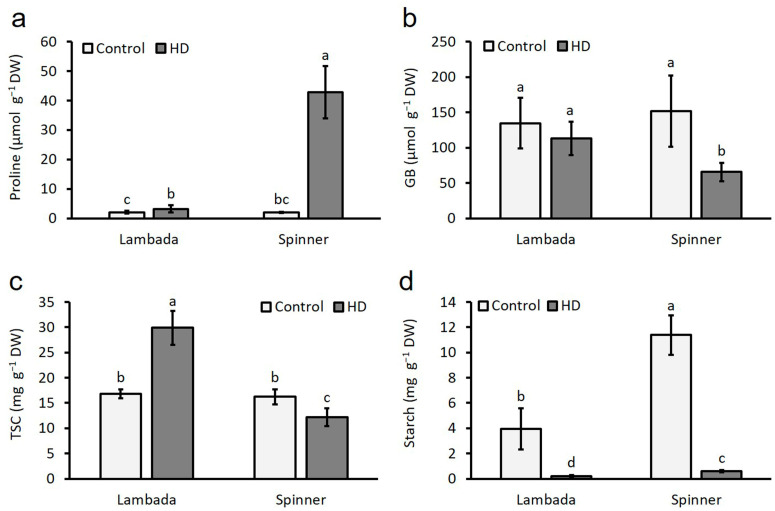
Accumulation of proline (**a**), glycine betaine (**b**), total soluble carbohydrates (**c**) and starch (**d**) in the flag leaves of the control and HD Lambada and Spinner plants. GB—glycine betaine, TSC—total soluble carbohydrates. Mean values, along with standard deviations, are presented. In each histogram, different letters above columns indicate significant difference of means minimum at the *p* ≤ 0.05 level of probability.

**Figure 8 plants-12-03907-f008:**
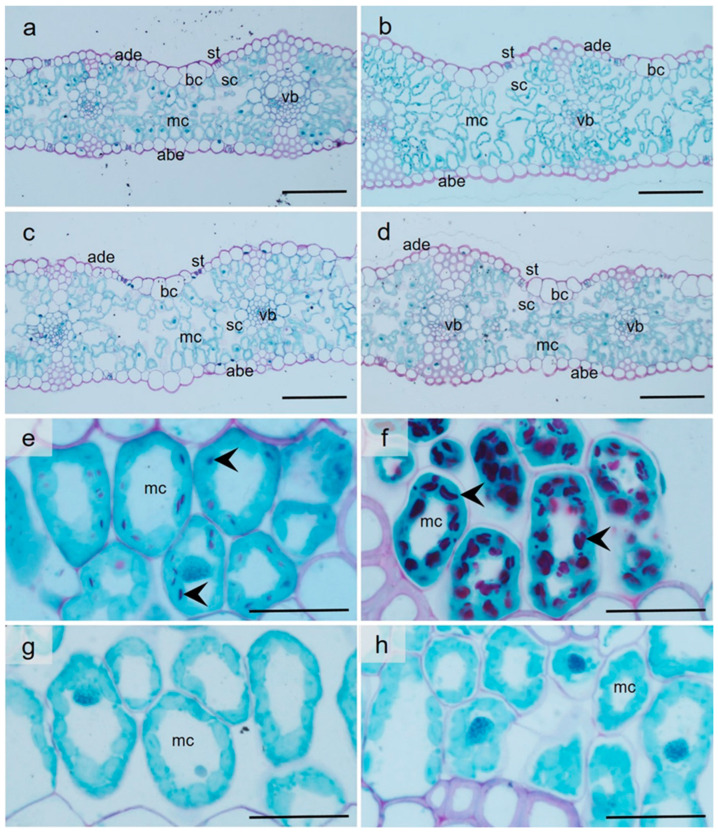
Structure of the cross-sectioned flag leaves (**a**–**d**) and starch distribution in the chloroplasts of the mesophyll cells (**e**–**h**) of the control (**a**,**b**,**e**,**f**) and heat and drought co-stressed (**c**,**d**,**g**,**h**) Lambada (**a**,**c**,**e**,**g**) and Spinner (**b**,**d**,**f**,**h**) plants (representative images). abe—abaxial epidermis, ade—adaxial epidermis, bc—bulliform cell, mc—mesophyll cell, sc—substomatal cavity, st—stoma, vb—vascular bundle, arrowhead—chloroplasts with primary starch granules, blue coloration—proteins, red coloration—carbohydrates. Scale bars: (**a**–**d**) = 100 µm; (**e**–**h**) = 30 µm.

**Figure 9 plants-12-03907-f009:**
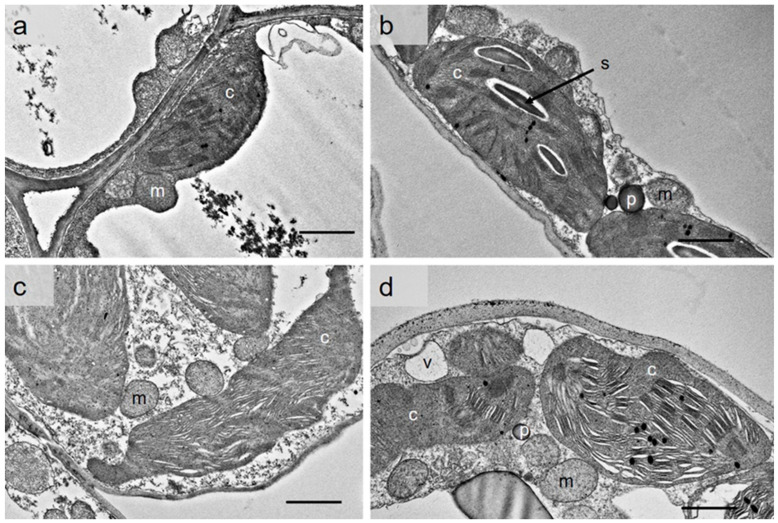
Ultrastructure of control (**a**,**b**) and heat and drought co-stressed (**c**,**d**) mesophyll cells of Lambada (**a**,**c**) and Spinner (**b**,**d**) flag leaves. c—chloroplast, m—mitochondrion, p—peroxisome, s—starch, v—vacuole. Scale bar = 1 µm.

**Table 1 plants-12-03907-t001:** Morphological traits of barley plants and anatomical features of flag leaves.

Trait	Lambada Control	Lambada HD Co-Stress	Spinner Control	Spinner HD Co-Stress
Plant height (cm)	70.5 ± 5.8 ^b^	62.1 ± 6.9 ^c^	92.5 ± 8.7 ^a^	72.9 ± 6.4 ^b^
Peduncle length (cm)	20.7 ± 3.1 ^b^	12.4 ± 4.5 ^c^	27.7 ± 4.0 ^a^	14.9 ± 4.6 ^c^
Leaf thickness (mm)	143.1 ± 13.0 ^b^	175.6 ± 12.0 ^a^	165.0 ± 19.6 ^a^	134.6 ± 7.9 ^b^
Bulliform CPA (µm^2^)	508.5 ± 68.2 ^ab^	547.7 ± 31.4 ^a^	449.7 ± 30.6 ^b^	397.5 ± 79.9 ^b^
Mesophyll CPA (µm^2^)	342.0 ± 85.5 ^a^	316.0 ± 106.8 ^a^	268.7 ± 52.1 ^b^	275.3 ± 116.9 ^b^
Chloroplast mesophyll CPA^−1^	6.9 ± 1.3 ^c^	7.1 ± 1.8 ^bc^	9.2 ± 1.4 ^a^	8.5 ± 1.6 ^ab^

Values represent means ± standard deviations. CPA—cell plan area; HD—heat and drought. Means in each row superscripted by different letters are significantly different minimum at the *p* ≤ 0.05 level of probability.

**Table 2 plants-12-03907-t002:** Pearson’s correlation coefficients (r) describing associations of physiological, biochemical, morphological and agronomic traits.

Traits	RWC	F_v_/F_m_	P_n_	g_s_	C_i_	E	WUE	Prot	Chl_a_	Chl_b_	Car	Chl_a/b_	Pro	GB	TSC	Starch	PH	PL	GPT	PP	TGW	HI
RWC	1																					
F_v_/F_m_	0.803 **	1																				
P_n_	0.793 **	0.916 **	1																			
g_s_	0.843 **	0.624	0.780 **	1																		
C_i_	0.498	0.329	0.592 *	0.756 **	1																	
E	0.684	0.747 **	0.925 **	0.816 **	0.835 **	1																
WUE	−0.105	−0.023	−0.282	−0.541	−0.833 **	−0.591 *	1															
Prot	0.824 **	0.634 *	0.647 *	0.854 **	0.651 *	0.672 *	−0.441 *	1														
Chl_a_	0.733 **	0.648 *	0.625 *	0.726 **	0.483	0.567	−0.335	0.960 **	1													
Chl_b_	0.757 **	0.690 *	0.673 *	0.747 **	0.507	0.613 *	−0.340	0.965 **	0.998 **	1												
Car	−0.730 **	−0.927 **	−0.950 **	−0.655 *	−0.444	−0.835 **	0.127	−0.562	−0.517	−0.573	1											
Chl_a/b_	0.592 *	0.437	0.373	0.567	0.316	0.309	−0.233	0.886 **	0.955 **	0.934 **	−0.268	1										
Pro	−0.781 **	−0.938 **	−0.825 **	−0.573	−0.127	−0.580 *	−0.136	−0.622 *	−0.672 *	−0.704 **	0.851 **	−0.516	1									
GB	0.891 **	0.768 **	0.728 **	0.827 **	0.477	0.632 *	−0.250	0.929 **	0.902 **	0.916 **	−0.685 *	0.806 **	−0.774 **	1								
TSC	0.062	0.368	0.066	−0.318	−0.750 **	−0.289	0.825 **	−0.224	−0.061	−0.052	−0.232	−0.052	−0.527	0.029	1							
Starch	0.735 **	0.467	0.500	0.797 **	0.630 *	0.540	−0.489	0.971 **	0.925 *	0.917 **	−0.382	0.902 **	−0.460	0.884 **	−0.335	1						
PH	0.415	0.162	0.165	0.514	0.596 *	0.325	−0.541	0.764 **	0.713 *	0.694 *	−0.080	0.754 **	−0.082	0.588 *	−0.507	0.856 **	1					
PL	0.638 *	0.485	0.549	0.756 **	0.770 **	0.667 *	−0.616 *	0.915 **	0.854 *	0.858 **	−0.445	0.785 **	−0.384	0.765 **	−0.441	0.920 **	0.908 **	1				
GPT	0.842 **	0.890 **	0.845 **	0.757 **	0.484	0.772 **	−0.271	0.710 **	0.671 *	0.706 *	−0.818 **	0.472	−0.814 **	0.836 **	0.121	0.609 *	0.318	0.575	1			
PP	0.759 **	0.927 **	0.985 **	0.727 **	0.542	0.905 **	−0.231	0.590 *	0.544	0.595 *	−0.949 **	0.282	−0.814 **	0.665 *	0.112	0.411	0.106	0.491	0.845 **	1		
TGW	0.628 *	0.685 *	0.792 **	0.709 **	0.686 *	0.799 **	−0.427	0.756 **	0.716 *	0.747 *	0.736 **	0.551	−0.551	0.732 **	−0.215	0.668 *	0.487	0.751 **	0.580 *	0.742 **	1	
HI	0.554	0.687 *	0.742 **	0.694 *	0.631 *	0.808 **	−0.629 *	0.612 *	0.572	0.605 *	−0.694 *	0.364	−0.565	0.669	−0.172	0.548	0.349	0.590 *	0.873 **	0.732 **	0.574	1

Car (carotenoids), Chl_a_ (chlorophyll *a*), Chl_b_ (chlorophyll *b*), Chl_a/b_ (chlorophyll *a* to *b* ratio), C_i_ (intercellular CO_2_ content), E (transpiration), F_v_/F_m_ (chlorophyll fluorescence parameter), GB (glycine betaine), GP (grain production), GPT (grain number per productive tiller), g_s_ (stomatal conductance), HI (harvest index), PH (plant height), PL (peduncle length), P_n_ (net photosynthetic rate), Pro (proline), Prot (total soluble protein content), RWC (relative water content), TGW (thousand-grain weight), TSC (total soluble carbohydrate content), WUE (water use efficiency), 1–0.9 (absolute magnitude (AM) of very strong correlation), 0.89–0.7 (AM of strong correlation), 0.69–0.4 (AM of moderate correlation), 0.39–0.1 (AM of weak correlation), 0.1–0 (AM of negligible correlation). * Correlation is significant at the 0.05 level of probability (two-tailed Pearson’s correlation test). ** Correlation is significant at the 0.01 level of probability (two-tailed Pearson’s correlation test).

## Data Availability

The data that support the findings of this study are available on request from the corresponding author. The data are not publicly available due to privacy or ethical restrictions.

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
