# Peer review of "Morpho-Anatomical, Physiological and Biochemical Adjustments in Response to Heat and Drought Co-Stress in Winter Barley"

_plants, 2023, doi:10.3390/plants12223907_

Round 1

Reviewer 1 Report

Comments and Suggestions for Authors

This study sought to show how two varieties of barley behaved in the face of combined drought and heat stress and for this purpose, a series of physiological, biochemical, morphological, and anatomical analyses were developed. The work is well structured and has the potential to be accepted for publication, however, I note below some points that should be improved before final acceptance:

Lines 101-104 - Neither of the genotypes showed signs of leaf rolling because of 101 HD co-stress Figure 1); however, flag leaves of the Spinner variety were rather wilted. 102 Moreover, HD caused yellowing and senescence of lower leaves in the Spinner variety 103 (Figure 1d). None of these characteristics are visible in the figures shown. Authors must provide better-quality figures or point out the location where the signs are being viewed. Furthermore, these figures require a scale. Should Figure 2 follow after its call in the text, not before

Line 146, how does the osmotic adjustment (ψw) have a positive sign? See that the explanatory text of how this variable was obtained (described in lines 644-651) does not seem to give rise to the fact that the stressed treatment presented an osmolality equivalent to 1 MPa more than its control (ψw = 0). It seems to me that the authors should review their calculations here.

Pearson correlations should not be presented in duplicate, but only half of the correlations. Furthermore, all abbreviations used in the table must be clearly indicated in the table

The experimental design of the study was bifactorial, as two varieties of barley (Lambada and Spinner) and two treatments (control and HD) were tested; however, the statistical analyses were carried out using One-Way ANOVA, disregarding the fact that there may have been an interaction between the factors, which requires showing the effect of the factors independently. So, unless the authors show me the Anova tables for all the statistical analyses, I am wary of approving a One-Way Anova analysis as if it were a Two-Way Anova analysis.

As significant as a 2% reduction in the Fv/Fm ratio is, this reduction does not appear to be physiological if not purely mathematical. The authors must make this clear, as they would be violating the fundamental laws of the chlorophyll a fluorescence measurement technique.

In figure 8F it is stated that the arrowheads represented starch, but in this case, I believe that they must be representing chloroplasts with primary starch inside them, which would still be an exaggeration. Authors should review the slides and make sure they are describing the correct structure.

Lines 450-452: “Higher thermal stability of cell compartments and molecules, such as PSII and photosynthetic pigments, resulted in sustained physiological and biochemical functioning and thus, higher plant production.” This sentence seems to come from a reference, the data presented in the manuscript is not enough to say this without the help of a reference.

Lines 455-457. “A reduced E in both barley genotypes, as observed in our study, could be linked to a reduced gs, similarly reported in maize [40], proposing that the effect of drought stress was prevalent or greater over heat stress” In this sentence the authors correlate gs with E, something really very pertinent, but immediately afterwards they correlate gs with heat stress. Even though gs correlates with E, gs per se does not show a direct correlation with heat stress

Analyzing the data in Figure 6, the authors must present a hypothesis that explains the decrease in gs with an increase in Ci in some phases and a decrease in Ci in other phases. In my opinion, this is one of the main results of the study and should be clarified.

Line 473. I invite authors to read recent articles on the role of proline, which reclassified the role of proline, now as a dissipator of reducing power, no longer as an osmoprotector. In this sense, the use of proline as an amino acid that promotes osmotic adjustment is wrong or minimally old and must be corrected. In this sense, the text encompassing lines 498 and 513 should be revised and only included in this space when the role of proline was attributed to better dissipation of reducing power in photosystems, no longer as an osmoprotective amino acid. Suggested reading https://doi.org/10.3390/su142315503, figure 10 and explanatory text.

Lines 526-538. I suggest that reviewers read more current texts on the topic and found that this topic has been widely discussed in several current scientific articles and today it is no longer thought the way it was thought in the 1987s when Pyke and Leech published their studies. Suggested reading https://doi.org/10.1111/nph.15730.

Minor revision

Keywords: exclude barley, heat, and drought co‐stress because these words are already in the tittle

There are a lot of long sentences that you finish reading and you don't even remember how they started. I suggest that authors make more succinct sentences with less than 400 characters including spaces (eg. Lines 64-71; Line 82-87 etc)

Line 129. What are VWC and RWC?

Line 146, Chance mPa by MPa

Item 2.5. The ordering of ideas makes the text tiring and confusing. Authors must count on the help of a native English speaker and make this text more fluid.

Lines 253-254. “Interestingly, proline contents were negatively correlated to all traits examined but Car”. This phrase seems to be incomplete

Line 334 “Fábián et al. (2019) and, Prasad et al. (2011) and Qaseem et al. (2019)” These references aren’t in Plants – MDPI rule. Line 363 Hossain and co-workers (2012) and line 365 too. A lot of references are listed in another rule different from Plants – MDPI. Double check.

Line 681. RPM is no longer an SI unit, this is due to the fact that to know the rotational force of a rotor having its values recorded in rpm, several factors must be considered, such as the diameter of the rotor. Without presenting the rotor diameter and other related considerations, the rotational force must be converted into g-force. Normally, all centrifuge catalogs come with this calculation ready, or on the centrifuge display you just need to change it to g form, and the value is automatically expressed in g force. In this sense, I suggest updating this value for g-force.

Comments on the Quality of English Language

The ordering of ideas makes the text tiring and confusing. Authors must count on the help of a native English speaker and make this text more fluid.

Author Response

Response Letter to the Reviewer #1

31 October 2023

Dear Referee,

Thank you very much for reading and revising our manuscript. Thank you for your comments.

Please find below all the changes made in the manuscript based on your review.

Comment/Suggestion 1: Lines 101-104 - Neither of the genotypes showed signs of leaf rolling because of 101 HD co-stress Figure 1); however, flag leaves of the Spinner variety were rather wilted. 102 Moreover, HD caused yellowing and senescence of lower leaves in the Spinner variety 103 (Figure 1d). None of these characteristics are visible in the figures shown. Authors must provide better-quality figures or point out the location where the signs are being viewed. Furthermore, these figures require a scale. Should Figure 2 follow after its call in the text, not before

Answer 1: Scale bars were added to Figure 1 and the location of leaf yellowing and senescence was pointed out in Figure 1d. Figure 2 has been replaced after its call in the text.

Comment/Suggestion 2: Line 146, how does the osmotic adjustment (ψw) have a positive sign? See that the explanatory text of how this variable was obtained (described in lines 644-651) does not seem to give rise to the fact that the stressed treatment presented an osmolality equivalent to 1 MPa more than its control (ψw = 0). It seems to me that the authors should review their calculations here.

Answer 2: Thank you very much for your comment. We reviewed the calculation of osmotic adjustment and found the mistakes made during the calculation. The revised Figure 3 and description of HD co-stressed induced alterations are included in the manuscript.

Comment/Suggestion 3: correlations. Furthermore, all abbreviations used in the table must be clearly indicated in the table

Answer 3: Table 2 was corrected, now it contains only half of the correlations, and all abbreviations used in the table are indicated in the table caption.

Comment/Suggestion 4: The experimental design of the study was bifactorial, as two varieties of barley (Lambada and Spinner) and two treatments (control and HD) were tested; however, the statistical analyses were carried out using One-Way ANOVA, disregarding the fact that there may have been an interaction between the factors, which requires showing the effect of the factors independently. So, unless the authors show me the Anova tables for all the statistical analyses, I am wary of approving a One-Way Anova analysis as if it were a Two-Way Anova analysis.

Answer 4: Thank you very much for your comment! 4.11 The Statistical analysis section of the manuscript was corrected as follows: All data were pooled means from the replicates, subjected to ANOVA (SSPS version 16.0, IBM Corp., Armonk, NY, USA), and the mean values were compared by Tukey’s multiple range test taking P ≤ 0.05 as significant to compare the differences between treatments and genotypes.  Pearson’s correlation coefficient was used to deduce the relationships between the measured traits (SSPS for Windows, version 16.0, IBM Corp., Armonk, NY, USA). Correlation coefficients (r) were interpreted according to Schober and co-workers (2018): 1–0.9 absolute magnitude (AM) of very strong correlation, 0.89–0.7 AM of strong correlation, 0.69–0.4 AM of moderate correlation, 0.39–0.1 AM of weak correlation, 0.1–0 AM of negligible correlation [102]. Mean values and standard deviations are presented in tables and figures. The standard deviation (SD) of means and differences between treatments were compared pairwise at P ≤ 0.05 level of probability.

Comment/Suggestion 5: As significant as a 2% reduction in the Fv/Fm ratio is, this reduction does not appear to be physiological if not purely mathematical. The authors must make this clear, as they would be violating the fundamental laws of the chlorophyll a fluorescence measurement technique.

Answer 5: The sentence “While the 5-day-long stress treatment triggered a significant (2%) reduction in the Fv/Fm ratio of the Spinner genotype, with a value of 0.78 as compared to its control counterpart, Lambada showed no significant modification in this parameter.” was replaced by “In our opinion, the statistically significant 2% decrease in the Fv/Fm ratio of the Spinner variety was not caused by the physiological changes triggered by the 5-day stress treatment, but by the low standard deviation of the measured values.”The relevant part has also been removed from the Discussion.

Comment/Suggestion 6: In figure 8F it is stated that the arrowheads represented starch, but in this case, I believe that they must be representing chloroplasts with primary starch inside them, which would still be an exaggeration. Authors should review the slides and make sure they are describing the correct structure.

Answer 6: We reviewed the sections and found that the arrowheads actually marked starch-containing chloroplasts according to the reviewer's opinion.

Comment/Suggestion 7: Lines 450-452: “Higher thermal stability of cell compartments and molecules, such as PSII and photosynthetic pigments, resulted in sustained physiological and biochemical functioning and thus, higher plant production.” This sentence seems to come from a reference, the data presented in the manuscript is not enough to say this without the help of a reference.

Answer 7: The sentence “Higher thermal stability of cell compartments and molecules, such as PSII and photosynthetic pigments, resulted in sustained physiological and biochemical functioning and thus, higher plant production.” has been removed from the manuscript.

Comment/Suggestion 8: Lines 455-457. “A reduced E in both barley genotypes, as observed in our study, could be linked to a reduced gs, similarly reported in maize [40], proposing that the effect of drought stress was prevalent or greater over heat stress” In this sentence the authors correlate gs with E, something really very pertinent, but immediately afterwards they correlate gs with heat stress. Even though gs correlates with E, gs per se does not show a direct correlation with heat stress

Answer 8: The sentence “A reduced E in both barley genotypes, as observed in our study, could be linked to a reduced gs, similarly reported in maize [40], proposing that the effect of drought stress was prevalent or greater over heat stress” was corrected as follows: “A reduced E in both barley genotypes, as observed in our study, could be linked to a reduced gs, similarly reported in maize [40].”

Comment/Suggestion 9: Analyzing the data in Figure 6, the authors must present a hypothesis that explains the decrease in gs with an increase in Ci in some phases and a decrease in Ci in other phases. In my opinion, this is one of the main results of the study and should be clarified.

Answer 9: A hypothesis/explanation was added to the Discussion section of the manuscript.

Comment/Suggestion 10: Line 473. I invite authors to read recent articles on the role of proline, which reclassified the role of proline, now as a dissipator of reducing power, no longer as an osmoprotector. In this sense, the use of proline as an amino acid that promotes osmotic adjustment is wrong or minimally old and must be corrected. In this sense, the text encompassing lines 498 and 513 should be revised and only included in this space when the role of proline was attributed to better dissipation of reducing power in photosystems, no longer as an osmoprotective amino acid. Suggested reading https://doi.org/10.3390/su142315503, figure 10 and explanatory text.

Answer 10: The role of proline has been corrected in the Discussion part of the manuscript. The old references were removed and more relevant references were added to the References:

Comment/Suggestion 11: Lines 526-538. I suggest that reviewers read more current texts on the topic and found that this topic has been widely discussed in several current scientific articles and today it is no longer thought the way it was thought in the 1987s when Pyke and Leech published their studies. Suggested reading https://doi.org/10.1111/nph.15730.

Answer 11: Thank you very much for your suggestion and the link to the outstanding literature on chloroplast ultrastructure in plants. The reference Pyke and Leech (1987) [85] was removed from the manuscript.

Minor revision

Comment/Suggestion 12: Keywords: exclude barley, heat, and drought co‐stress because these words are already in the tittle

Answer 12: Barley, heat, and drought co‐stress were excluded from the Keywords.

Comment/Suggestion 13: There are a lot of long sentences that you finish reading and you don't even remember how they started. I suggest that authors make more succinct sentences with less than 400 characters including spaces (eg. Lines 64-71; Line 82-87 etc)

Answer 13: The number of long sentences was reduced.

Comment/Suggestion 14: Line 129. What are VWC and RWC?

Answer 14: Meanings of VWC—volumetric water content (VWC) and RWC—relative water content were added to the Results part of the manuscript.

Comment/Suggestion 15: Line 146, Chance mPa by MPa

Answer 15: mPa was changed to MPa.

Comment/Suggestion 16: Item 2.5. The ordering of ideas makes the text tiring and confusing. Authors must count on the help of a native English speaker and make this text more fluid.

Answer 16: Unfortunately, during the editing of the manuscript, the Table 2 caption was inserted into this section that originally sounded as follows:

“2.5 Heat and drought co-stress reduced total soluble protein and photosynthetic pigment contents

The amount of total soluble proteins in the control flag leaves of both Lambada (33.4 mg g-1 DW) and Spinner (48.4 mg g-1 DW) genotypes was significantly reduced due to the applied HD treatment by 31% and 63%, respectively, as compared to their respective control counterparts (Figure 4a). The concentration of the photosynthetic pigments was measured in terms of the Chla, Chlb, Chla+b and Car contents in the flag leaves and the ratios of Chla/Chlb and Car/Chl were calculated. The chlorophyll contents of the control plants varied with the genotypes. Compared to the total pigment contents of the Spinner flag leaves (10.1 mg g-1 DW), those of Lambada (16.8 mg g-1 DW) were significantly lower under control conditions (Figure 4b-d). On a dry weight basis, Chla, Chlb and Chla+b contents underwent a significant decrease under HD co-stress conditions in both Lambada and Spinner flag leaves by 16%, 19% and 16% and 60%, 62% and 61%, respectively. Based on the acquired values, the reduction in these pigments was much more substantial in the plants of the Spinner genotype, the lower leaves of which turned yellow at the end of the 5-day treatment (Figure 1d). Nevertheless, the decrease in HD-induced total Chl contents (˗16%, ˗60%) was less than the decrease in Pn (˗46%, ˗80%). HD co-stress resulted in a 5% non-significant increase in Car contents of Lambada flag leaves despite inducing a 53% significant decline in the Spinner variety (Figure 4e). HD co-stress significantly increased the chlorophyll a/b ratio in the flag leaves of both Lambada and Spinner by 4% and 6%, respectively (Figure 4f). The Car/Chl ratio was increased by a significant 21% and 15% in Lambada and Spinner varieties, respectively (Figure 4g). The HD-induced reduction of Chla and Chlb contents was in a strong positive correlation with the decline of the plant RWC (Table 2).”

Comment/Suggestion 17: Lines 253-254. “Interestingly, proline contents were negatively correlated to all traits examined but Car”. This phrase seems to be incomplete

Interestingly, proline contents were negatively correlated to all traits examined but Car

Answer 17: The phrase was corrected as follows: “Interestingly, proline contents were negatively correlated to all traits examined but carotenoid contents.”

Comment/Suggestion 18: Line 334 “Fábián et al. (2019) and, Prasad et al. (2011) and Qaseem et al. (2019)” These references aren’t in Plants – MDPI rule. Line 363 Hossain and co-workers (2012) and line 365 too. A lot of references are listed in another rule different from Plants – MDPI. Double check.

Answer 18: The references were corrected.

Comment/Suggestion 19: Line 681. RPM is no longer an SI unit, this is due to the fact that to know the rotational force of a rotor having its values recorded in rpm, several factors must be considered, such as the diameter of the rotor. Without presenting the rotor diameter and other related considerations, the rotational force must be converted into g-force. Normally, all centrifuge catalogs come with this calculation ready, or on the centrifuge display you just need to change it to g form, and the value is automatically expressed in g force. In this sense, I suggest updating this value for g-force.

Answer 19: The RPM was updated to g-value.

Yours sincerely,

Dr. Katalin Jäger

Reviewer 2 Report

Comments and Suggestions for Authors

General comments

I have read the manuscript:  Plants MDPI. Entitle: Morpho‐anatomical, physiological, and biochemical adjustments in response to heat and drought co‐stress in two‐rowed winter barley genotypes with contrasting tolerance written byEmmanuel Asante Jampoh et. al., for publication of Plant MDPI. In this study, author investigate the combined effect of heat and drought stress of winter barley genotype with contrasting tolerance. Author investigates the barley genotype phenology, morpho-anatomy, physiological and biochemical responses, and yield constituents. Author mainly found that the stress induced reduction in relative water content, osmotic adjustment, total soluble protein and carbohydrate contents, photosynthetic pigment and photosynthetic efficiency.

The overall research is well conducted and very information because of included many traits and its depth study but author should further improvement of this manuscript for journal acceptance. Research is obvious application potential for the readers because this research provides the important finding of greater tolerance barley genotype, Lambada variety with relative resistance of photosynthetic pigments towards stress triggered degradation, retained photosynthetic parameters and ultra structure. In this sense, this manuscript is much more valuable. However, I found a lack of story connection and lack of potential references (some I suggested some below). Overall after I evaluate and request the author for this manuscript as a “MAJOR REVISION”.

 Major Suggestions

1) Abstract: Author should improve abstract further. Abstract should be more informative; it should present less in methodology part and include clear result and common message for audiences. This is very important than only lengthening the text. Author should work in novelty; how does the study elucidate your finding useful for the society, please revised accordingly.

2) Hypothesis of the Review: Author well presented the hypothesis of the study in Ln. 77-79 in this paper. Please also focus the objectives of the study in the last section of the introduction. Author mention hypothesis and connect with the research objectives of the introduction section. The hypothesis and research objectives should be very clear because, without appropriate literature, questions, or hypotheses in the introduction section the entire text will be unclear.

3) Concise the text: author should concise the text by removing the unnecessary and less important text. Please include the text related on the research title and its circumstances by cutback unnecessary text.

Some line-to-Line comments

4) Line no. 37-40 (Introduction):  The whole introduction is well connected. However, author should be further clarifying the Line no. 37-40 by include the text related to the drought “reduction of plant morphology (reduced leaf size and stem length, leaf length/width, and vegetative growth) and physiological traits (reduction of photosynthesis, leaf water potential, and sap movement)”. These articles help to further clarify your introduction (1) https://doi.org/10.1016/j.scienta.2023.112276 (2) https://doi.org/10.1093/treephys/tpy153. Refer these articles as a reference.

 5) Line no. 328 (Discussion): Author should concise the text in the discussion section. Your discussion is very long, please cut back the length and unrelated text in this section by including the appropriate references.

 6) Line no. 405-410: author should further improve the Line no. 405-410 include the below reference for clear interpretation the text. The leaf anatomy specially its internal structure is very important for the photosynthesis efficiency. Refer this article as a reference. DOI:10.1016/j.scienta.2018.11.021 “Increase the palisade mesophyll parenchyma enhance the photosynthesis rate because its help to better capture the light and easily assist to change the light energy to chemical energy while performing the photosynthesis”. 

 7) Line no. 328 (Discussion): Author should mention main theme of antioxidant and secondary metabolites under drought/heat stress and address why ROS is emerging in stress conditions? Refer to these two articles for better clarify (1) https://doi.org/10.1038/s41598-019-55889 (2) https://doi.org/10.1016/j.scitotenv.2021.146466 and mention somewhere in that paragraph “abiotic stress especially environmental stress (e.g. UV radiation) plant produces the ROS when the plant exposed to the stress condition and plant produce antioxidant, flavonoids, and secondary metabolites , play to the role for protecting the plant for detoxifying ROS and protect the plant to protect from drought and help to stabilization of proline and amino acid”.

8) Conclusion: I did not see the conclusion section separately made by author. Please remember that conclusion should not be repetitive in the abstract or a summary of the results section. I would love to read striking points and take-home messages that will linger in the readers’ minds. What is the novelty, how does the study elucidate some questions in this field, and the contributions the paper may offer to the scientific community?

9) Line no. 745 (References): please include more related citation, check their pattern and writing style, spell check, and other grammatical errors. moreover, the author should cut the old and less matching literature and include the latest literature some of them are above.

Good Luck !

Author Response

Response Letter to the Reviewer #2

31 October 2023

Dear Referee,

Thank you very much for reading and revising our manuscript. Thank you for your comments.

Please find below all the changes made in the manuscript based on your review.

  1. Suggestion 1: Abstract: Author should improve abstract further. Abstract should be more informative; it should present less in methodology part and include clear result and common message for audiences. This is very important than only lengthening the text. Author should work in novelty; how does the study elucidate your finding useful for the society, please revised accordingly.

Answer 1: The Abstract was revised according to the Reviewer's request. The methodology part was cut and the usefulness of research for society was emphasized.

  1. Suggestion 2: Hypothesis of the Review: Author well presented the hypothesis of the study in Ln. 77-79 in this paper. Please also focus the objectives of the study in the last section of the introduction. Author mention hypothesis and connect with the research objectives of the introduction section. The hypothesis and research objectives should be very clear because, without appropriate literature, questions, or hypotheses in the introduction section the entire text will be unclear.

Answer 2: The objectives were made clearer and were connected with the hypothesis:

  1. Suggestion 3: Concise the text: author should concise the text by removing the unnecessary and less important text. Please include the text related on the research title and its circumstances by cutback unnecessary text.

Answer 3: The title and the text were shortened according to Reviewer’s suggestion, and the long sentences were made more succinct. The new title of the manuscript is “Morpho‐anatomical, physiological and biochemical adjustments in response to heat and drought co-stress in winter barley”.

  1. Comment 1: Line no. 37-40 (Introduction): The whole introduction is well connected. However, author should be further clarifying the Line no. 37-40 by include the text related to the drought “reduction of plant morphology (reduced leaf size and stem length, leaf length/width, and vegetative growth) and physiological traits (reduction of photosynthesis, leaf water potential, and sap movement)”. These articles help to further clarify your introduction (1) https://doi.org/10.1016/j.scienta.2023.112276 (2) https://doi.org/10.1093/treephys/tpy153. Refer these articles as a reference.

Answer 4: The text “Drought triggers the reduction of plant morphology and physiological traits” and the reference https://doi.org/10.1016/j.scienta.2023.112276 were added to the Introduction and References parts of the manuscript, respectively. As the reference Salmon et al. 2019 is a review and discusses the effect of drought on tree phloem, we hardly considered it relevant to the topic of the manuscript. We would like to highlight that the relevant topic is discussed extensively focusing on Graminae in the Discussion section.

  1. Comment 2: Line no. 328 (Discussion): Author should concise the text in the discussion section. Your discussion is very long, please cut back the length and unrelated text in this section by including the appropriate references.

Answer 5: The Discussion was shortened and the appropriate references were removed.

  1. Comment 3: Line no. 405-410: author should further improve the Line no. 405-410 include the below reference for clear interpretation the text. The leaf anatomy specially its internal structure is very important for the photosynthesis efficiency. Refer this article as a reference. DOI:10.1016/j.scienta.2018.11.021 “Increase the palisade mesophyll parenchyma enhance the photosynthesis rate because its help to better capture the light and easily assist to change the light energy to chemical energy while performing the photosynthesis”.:

Answer 6: The publication of Bhusal et al. 2019 was included to the Discussion and References sections.

  1. Comment 4: Line no. 328 (Discussion): Author should mention main theme of antioxidant and secondary metabolites under drought/heat stress and address why ROS is emerging in stress conditions? Refer to these two articles for better clarify (1) https://doi.org/10.1038/s41598-019-55889 (2) https://doi.org/10.1016/j.scitotenv.2021.146466 and mention somewhere in that paragraph “abiotic stress especially environmental stress (e.g. UV radiation) plant produces the ROS when the plant exposed to the stress condition and plant produce antioxidant, flavonoids, and secondary metabolites , play to the role for protecting the plant for detoxifying ROS and protect the plant to protect from drought and help to stabilization of proline and amino acid”.

 Answer 7: Paragraph on ROS was added the Discussion chapter of the study. Publication of Bhusal et al. (2021) was included into the Introduction and Discussion sections as it evaluates morphological, physiological, and biochemical traits for assessing drought resistance.

  1. Comment 5: Conclusion: I did not see the conclusion section separately madse by author. Please remember that conclusion should not be repetitive in the abstract or a summary of the results section. I would love to read striking points and take-home messages that will linger in the readers’ minds. What is the novelty, how does the study elucidate some questions in this field, and the contributions the paper may offer to the scientific community?

 Answer 8: According to the Instructions for Authors of the journal Plants, the section Conclusions is not mandatory. However, the main findings related to the research and the questions raised based on the results are summarized in the last paragraph of the Discussion section.

Suggestion 9: Line no. 745 (References): please include more related citation, check their pattern and writing style, spell check, and other grammatical errors. moreover, the author should cut the old and less matching literature and include the latest literature some of them are above.

Answer 9: References Bhusal et al. 1019, 2021 and 2023 were added to References

Yours sincerely,

Dr. Katalin Jäger

Round 2

Reviewer 2 Report

Comments and Suggestions for Authors

Dear Author

I have read the revised manuscript plants-2682990. Entitled: Morpho‐anatomical, physiological and biochemical adjustments in response to heat and drought co-stress in two-rowed winter barley genotypes with contrasting tolerance in plant MDPI. This is the second submission made by the author. The author addressed all the questions and suggestions that I raised the issue in the review of the original manuscript. I satisfy the author’s revisions. Author improved the abstract. Author significantly improved their research hypothesis and well connected with the research objectives in this time. This manuscript improved the flow of writing, which was comparatively shallow in the original version but in this revised copy author very well addressed all the quarries and suggestions. Before accepting this manuscript, please check again the referencing. Further if there is anything needed to be revised by the author, especially English grammar, or spell check, I request this manuscript is currently in “Minor Revision” and the author may correct any further grammatical errors (if any) the author may improve in this stage.

Thank you.

Author Response

Response Letter to the Reviewer #2

9 November 2023

Dear Referee,

Thank you very much for reading and revising our manuscript. Thank you for your comments.

Please find below all the changes made in the manuscript based on your review.

Comment of the Reviewer #2: Before accepting this manuscript, please check again the referencing. Further if there is anything needed to be revised by the author, especially English grammar, or spell check, I request this manuscript is currently in “Minor Revision” and the author may correct any further grammatical errors (if any) the author may improve in this stage.

Answer to the Reviewer #2: We have corrected errors in the manuscript. We have checked and corrected the references. The corrections made are indicated in red in the manuscript. The manuscript has been checked by an English language proofreader.

We hope that we have adapted the manuscript in accordance with your comments.

Yours sincerely,

Dr. Katalin Jäger
